# New Organoselenium (NSAIDs-Selenourea and Isoselenocyanate) Derivatives as Potential Antiproliferative Agents: Synthesis, Biological Evaluation and in Silico Calculations

**DOI:** 10.3390/molecules27144328

**Published:** 2022-07-06

**Authors:** Yousong Nie, Shaolei Li, Ying Lu, Min Zhong, Xiaolong Li, Youhong Zhang, Xianran He

**Affiliations:** 1School of Environmental Ecology and Biological Engineering, Wuhan Institute of Technology, LiuFang Campus, Guanggu 1st Road, Wuhan 430205, China; yousongnie@163.com; 2Shenzhen Fushan Biological Technology Co., Ltd., Kexing Science Park A1 1005, Nanshan Zone, Shenzhen 518057, China; lsl1917@163.com (S.L.); lixiaolong@szfs01.com (X.L.); 3Key Laboratory of Optoelectronic Chemical Materials and Devices of Ministry of Education, Jianghan University, Wuhan Economic and Technological Development Zone, Wuhan 430056, China; 202130773036@stu.jhun.edu.cn (Y.L.); 474488728@163.com (M.Z.); 4School of Medicine, Jianghan University, Wuhan Economic and Technological Development Zone, Wuhan 430056, China

**Keywords:** NSAIDs, isoselenocyanate, selenourea, antiproliferative

## Abstract

In this study, we report on the synthesis of new organoselenium derivatives, including nonsteroidal anti-inflammatory drugs (NSAIDs) scaffolds and Se functionalities (isoselenocyanate and selenourea), which were evaluated against four types of cancer cell line: SW480 (human colon adenocarcinoma cells), HeLa (human cervical cancer cells), A549 (human lung carcinoma cells), MCF-7 (human breast adenocarcinoma cells). Among these compounds, most of the investigated compounds reduced the viability of different cancer cell lines. The most promising compound **6b** showed IC_50_ values under 10 μM against the four cancer cell lines, particularly to HeLa and MCF-7, with IC_50_ values of 2.3 and 2.5 μM, respectively. Furthermore, two compounds, **6b** and **6f,** were selected to investigate their ability to induce apoptosis in MCF-7 cells via modulation of the expression of anti-apoptotic Bcl-2 protein, pro-inflammatory cytokines (IL-2) and proapoptotic caspase-3 protein. The redox properties of the NSAIDs-Se derivatives were conducted by 2, 2-didiphenyl-1-picrylhydrazyl (DPPH), bleomycin-dependent DNA damage and glutathione peroxidase (GPx)-like assays. Finally, a molecular docking study revealed that an interaction with the active site of thioredoxin reductase 1 (TrxR1) predicted the antiproliferative activity of the synthesized candidates. Overall, these results could serve as a promising launch point for further designs of NSAIDs-Se derivatives as potential antiproliferative agents.

## 1. Introduction

Non-steroidal anti-inflammatory drugs (NSAIDs) are a class of active pharmaceutical ingredients (API) that are widely used in the treatment of inflammatory conditions, including pain associated with arthritis, worldwide [1,2]. Since the 1980s, when the antiproliferative effect of sulindac was first observed in reducing colon adenomas [3], a growing body of studies has evaluated the chemoprevention and antiproliferative potential of NSAIDs [4,5,6], although their exact molecular mechanism has remained elusive.

The selenium (Se) has been recognized to play an important role in human health and disease [7,8]. In recent years, a number of studies have indicated an inverse association between Se intake and cancer risk [9,10,11]. There are three main categories of Se-containing compounds (inorganic, organic, and selenoproteins) with potential pharmacological properties; the most developed and studied are organoselenium compounds, which have been shown to inhibit the initiation and postinitiation phases of chemical carcinogenesis [12]. Organic selenium compounds with diverse functional groups, including selenoesters, methylseleninic acid, isoselenocyanates, diselenides, endocyclic selenium, selenocyanates and trifluoromethyl selenides, were found to show potent antiproliferative activity [13,14,15,16,17,18] (Figure 1).

Considering the chemo-preventive effects of NSAIDs and the antiproliferative activity of organic selenium compounds, along with reports that support the modification of NSAIDs scaffolds with Se functionalities [19,20] and our continued work to search for organoselenium derivatives with antiproliferative activity [21,22,23,24], several new NSAIDs-based derivatives, combining the NSAIDs scaffold with an organoselenium motif (Isoselenocyanate and selenoureas), were synthesized in this report (Figure 2). Their antiproliferative activities against SW480, HeLa, A549 and MCF-7 cell lines were shown using the MTT (3-(4,5-dimethylthiazol-2-yl)-2,5-diphenyltetrazolium bromide) assay. Two compounds, **6b** and **6f,** were selected to test the protein expression levels of Bcl-2, IL-2 and caspase-3 biomarkers in MCF-7 cells. Furthermore, the antioxidant potential of the compounds was investigated by employing DPPH, bleomycin-dependent DNA damage and GPx-like assays. Finally, Thioredoxin Reductase (TrxR1) was selected as a docking protein to predict the target and antiproliferative activity of the prepared NSAIDs-Se hybrid compounds.

## 2. Results and Discussion

### 2.1. Chemistry

The synthesis of novel families of NSAIDs-based seleno derivatives as potential antiproliferative agents: Isoselenocyanates (ISCs) (**5a**–**5h**) and selenoureas (**6a**–**6h**) in this study was performed as outlined in Figure 1.

The synthesis of the NSAIDs-Isoselenocyanate derivatives (**5a**–**5h**) (Purity ≥ 95% by HPLC) was started from 4-aminobenzylamine protected by bis(1,1-dimethylethyl) ester to afford compound **1**. Compound **1** was transformed into formamide **2** upon treatment with formic acid, with 80% yield. Next, compound **3** was obtained by deprotecting Boc-group in trifluoroacetic acid (TFA) and Dichloromethane (DCM) system. Compounds **4a**–**4h** (Purity ≥ 95% by HPLC, except 4g) were obtained by commercially available NSAIDs and compound **3** in the presence of EDCI and HOBT as condensantion agent, in DMF as sovent and under a nitrogen atmosphere. Then, compounds **4a**–**4h** were transformed into the NSAIDs-based isoselenocyanate (**5a**–**5h**) (Figure 1) in a one-pot two-step procedure, which involves triphosgene-based dehydration of **4a**–**4h** into a transient non-isolated isocyanide, followed by the addition of elemental selenium black (70% overall yield).

NSAIDs-selenourea derivatives (**6a**–**6h**) (Purity ≥ 95% by HPLC) were obtained by conducting corresponding NSAIDs-Isoselenocyanate derivatives with methanol.

### 2.2. Cell Viability Assay

MTT assay was conducted to evaluate the potential antiproliferative activities against human tumor cell lines derived from various human cancer types: SW480 (human colon adenocarcinoma cells), HeLa (human cervical cancer cells), A549 (human lung carcinoma cells), MCF-7 (human breast adenocarcinoma cells) of target compounds **5a**–**5h** and **6a**–**6h**, 5-Fu was selected as reference standard (Table 1).

Overall, the IC_50_ values obtained and summarized in Table 1 show that allfthe tested organoselenium compounds with NSAIDs scaffolds and Se functionalities (isoselenocyanate and selenourea) exhibit growth inhibition in all cancer cell lines, while the selected patent NSAIDs (Aspirin, Ibuprofen and Naproxen) are inactive against all cells, even at the maximum dose of 50 μM.

An overview analysis of the IC_50_ values obtained and summarized in Table 1 showed that most of the NSAIDs-selenourea derivatives wer emore effective than NSAIDs-isoselenocyanate derivatives against all four cancer cell lines. Furthermore, the most active compounds of these two series were **6b** and **6f**. These two compounds show IC_50_ values below 10 μM in all tested cancer cell lines. Compound **6b** was the most potent agent, with IC_50_ values below 5 μM in all cancer cell lines and remarkable antiproliferative activity against HeLa (2.3 μM) and MCF-7 (2.5 μM).

### 2.3. Evaluation of Bcl-2, IL-2 and Caspase-3 Molecular Biomarkers in MCF-7 Cells

To further understand the possible mechanism of the reduced cell viability of the target compounds, **6b** and **6f** were selected for their ability to induce apoptosis in MCF-7 cells via modulation of the expression of some apoptosis-related proteins (e.g., Bcl-2, IL-2, caspase-3). Protein levels of the anti-apoptotic marker Bcl-2 were measured using enzyme-linked immunosorbent assay (ELISA) according to the manufacturers’ instructions (Merck, Rahway, NJ, USA). Enzyme-linked immunosorbent assay was used for the quantitative detection of IL-2 and caspase-3 (Platinum ELISA).

As shown in Table 2, **6b** and **6f** were able to downregulate the expression of Bcl-2 and upregulate the expression of IL-2 and Caspase-3 in MCF-7 cells compared with untreated cells. Compound **6f** downregulated over 50% of the expression levels of Bcl-2 compared to untreated cells. Furthermore, compound **6b** modulated the IL-2 level to a 1.5-fold increase in expression when compared to the untreated control cells. Finally, compound **6b** exhibited a superior activity increased the expression level of caspase-3 by 5-fold compared to untreated cells. From these results, it is likely that compounds **6b** and **6f** may induce apoptosis to inhibit tumor cells growth, in line with the underlying mechanism of some organoselenium compounds, which were reported to be effective against prostate and oral carcinoma cells via the estimation of potential biomarkers [15].

### 2.4. Antioxidant Assay

Reactive oxygen species (ROS), e.g., H_2_O_2_, O_2_^−^, HO˙, and HOCl, serve important physiological roles that include signaling functions, host defense, and oxidative biosynthesis [25].

Various human diseases, including different types of cancer, are associated with a disturbed intracellular redox balance and oxidative stress (OS) [26]. Redox modulators play an important role as chemotherapeutic potential antitumor agents [27].

As a number of synthetic organoselenium compounds have been synthesized for their use as redox-modulators in recent years [28,29], the antioxidant activity of the selected synthesized compounds are further estimated by employing different biochemical assays such as DPPH, bleomycin-dependent DNA damage and Gpx-like assays [30,31,32].

#### 2.4.1. Radical Scavenging Capacity (DPPH) Assay

The DPPH scavenging capacity assay is considered a valid and rapid colorimetric method for antioxidant property evaluation. This assay has been successfully utilized for investigating the antioxidant properties of nutritional products and organic selenides [33,34].

As depicted in Table 3, NSAIDs-selenourea derivatives **6b**, **6f** and **6h** were the most active compounds in this assay, demonstrating a good free-radical scavenging activity compared to Vitamin C. The family of NSAIDs-selenourea derivatives is better than the corresponding NSAIDs-isoselenocyanate derivatives on this assay, except when comparing **5d** and **6d**.

#### 2.4.2. Bleomycin DNA Damage Assay

Bleomycin (BLM) is a family of glycopeptide antibiotics that are routinely used as antitumor agents; it is believed to oxidize DNA and induces single- and double-strand breaks [35,36]. The bleomycin-iron DNA damage assay has routinely been used as a preliminary method to test the potential of drugs and organic selenium compounds [37]. As shown in Table 3, compounds **5d**, **5h** and **6g** induced DNA degradation significantly more than other tested compounds.

#### 2.4.3. Glutathione Peroxidase-like Activity Assay

Glutathione peroxidase (GPx) is a well-known selenoenzyme that functions as an antioxidant [38,39]. The potential antioxidant activity of all the NSAIDs-Se derivatives were estimated using an NADPH-reductase coupled assay [40,41]. The GPx activity of the synthesized compounds was estimated by the decrease in absorbance (340 nm) due to the oxidation of NADPH to NADP^+^. Ebselen was used as a positive control.

As shown in Table 4, compounds **6a**–**6h** displayed a better GPx-like activity than the respective **5a**–**5h** derivatives. Compound **6g** was the most active derivative in this assay, reaching up to 1.5-fold of the GPx mimetic ebselen.

### 2.5. Docking Studies

TrxR1 consists of four monimers, which have FAD and NAD binding domains at the N-terminal and the dimerization interface domain at the flexible C-terminal side. The binding mode between organoselenium compounds and Mammalian TrxR1 protein was described by docking studies. TrxR1 consists of several functional domains, including FAD- and NAD-binding domains at the N-terminal, and the dimerization interface domain at the flexible C-terminal side [42,43,44]. It has been reported that flexible docking can simulate the interaction between small molecules and TrxR1 [45]. Therefore, compounds **6b** and **6f** were docked into the TrxR1 protein (PDB id: 1H6V) using the Flexible Docking Protocol, as reported in the literature [45]. The distances between the selenium atom of all two compounds and Cys497/Cys498 of TrxR1 were measured and focused on because they are closely related to the accessibility of cysteine thiol attack selenides.

The pose 2 of compound **6b** showed high -CDOCKER energy values (21.223 kcal/mol) and a close distance between the selenium atom and Cys498 (7.489 Å). Multiple interactions were found in this conformation, including electrostatic (Pi-Anion) between the benzene and GLU477 (4.326 Å), hydrogen bonding between the hydrogen of secondary amine of Seleno-carbamates and GLU477 (2.097 Å), and hydrogen bonding between the hydrogen of hydroxyl of Seleno-carbamates and GLU477 (2.549 Å) (Table 5, Pose 2; Figure 3). Compound **6f** had multiple interactions with GLY496, which was adjacent to the key amino acid Cys497/Cys498. In pose 1, the value of -CDOCKER energy was 33.662 kcal/mol and distance between the selenium atom and Cys498 was 5.167 Å (Table 6, Pose 1). Three hydrogen bondings appear in this conformation, including the hydrogen bonding between the nitrogen of selenocarbamates and GLU477 (2.834 Å), the hydrogen bonding between the hydrogen of amide bond and GLY496 (2.776 Å) and the hydrogen bonding between the oxygen of amide bond and GLY496 (2.494 Å) (Figure 4).

## 3. Conclusions

In conclusion, the present study involved the synthesis of new organoselenium derivatives, including NSAIDs scaffolds and Se functionalities (isoselenocyanate and selenourea). Compound **6b** exhibited the most potent activity in the MTT assay, with remarkable antiproliferative activity against HeLa (IC_50_ = 2.3 μM) and MCF-7 (IC_50_ = 2.5 μM). Compounds **6b** and **6f** were selected to verify if organic selenides can induce apoptosis in MCF-7 cells by modulating the expression of the Bcl-2, IL-2 and caspase-3 molecular biomarkers. The selected compounds were able to downregulate the expression of Bcl-2 and upregulate the expression of IL-2 and Caspase-3 in MCF-7 cells compared with untreated cells. Furthermore, most of the synthesized NSAIDs-Se hybrid compounds exhibited antioxidant activity in antioxidant evaluation, including DPPH, bleomycin-dependent DNA damage and Gpx-like assays.

Finally, in a flexible docking study performed on the TrxR1 enzyme, compound **6b** showed promising binding energies and a promising binding mode for the distance between the selenium atom and Cys497/Cys498. At this point, compound **6b** may act as a TrxR inhibitor. Further investigations of antiproliferative candidates based on these results are in progress.

## 4. Materials and Methods

### 4.1. Materials

All chemical reagents for the synthesis of the compounds were purchased from Macklin (Shanghai, China) or TCI (Shanghai, China) and used without further purification unless stated otherwise. TLCs were performed on aluminium pre-coated sheets (E. Merck Silica gel 60 F254). Melting points (uncorrected) were recorded on an Electrothermal apparatus. ^1^H (400 MHz), ^13^C (100 MHz) NMR and ^19^F (376 MHz) spectra were recorded at 25 °C on a Bruker Avance 400 MHz spectrometer with 5-mm PABBO probe. Chemical shifts (δ) are reported in parts per million (ppm) and the coupling constants (*J*) are expressed in Hertz (Hz). Mass analysis was recorded on an ESI source mass detector (Thermo LCQ FLEET).

### 4.2. Experimental Procedures

#### 4.2.1. N-(4-(aminomethyl)phenyl)formamide 3

Trifluoroacetic acid (TFA, 1.2 eq.) was added to a solution of compound **2** (1.0 eq.) in DCM (5 mL) at −10 °C. The mixture was stirred at −10 °C for 1 h. TLC showed the reaction was complete. The mixture was concentrated under reduced pressure. The crude was diluted with DCM (10 mL) and the PH was adjusted to 8~9 by TEA. The mixture was used in the next step without purification.

#### 4.2.2. General Procedure for the Synthesis of Compounds **4a**–**4h**

1-ethyl-3(3-dimethylpropylamine) carbodiimide (EDCI, 1.2 eq.), 1-hydroxybenzotriazole (HOBT, 1.2 eq.) and TEA (3.0 eq.) was added to a solution of patent NSAIDS (1.o eq.) in DCM (5 mL) and DMF (5 mL). The mixture was stirred at 25 °C for 30 min under nitrogen atmosphere. Then, N-(4-(aminomethyl)phenyl)formamide (compound **3**, 1.2 eq.) was added to the mixture. The mixture was stirred at 25 °C for 16 h under inert atmosphere. TLC showed the reaction was complete. The mixture was diluted with H_2_O (20 mL); the aqueous layer was extracted with DCM (15 mL × 2); the combined organic layer was washed with brine (20 mL × 3), dried over Na_2_SO_4_ and filtered, and the filtrate was concentrated under reduced pressure. The residue was purified by column chromatography on silica gel, eluting with dichloromethane/methanol solution to obtain the desired compound.

##### *N*-(4-formamidobenzyl)-2-(4-isobutylphenyl)propenamide (**4a**)

Yield: 68%. White solid. Mp: 87–89 °C. ^1^H NMR (400 MHz, CDCl_3_): δ 0.86 (d, *J* = 8.00 Hz, 6H, 2 × -CH_3_), 1.34 (d, *J* = 8.00 Hz, 3H, -CH_3_), 1.77–1.84 (m, 1H, -CH), 2.41 (d, *J* = 8.00 Hz, 2H, -CH_2_), 3.63 (q, *J* = 8.00 Hz, 1H, -CH), 4.18–4.20 (m, 2H, -CH_2_), 7.06–7.09 (m, 4H, Ar-H), 7.23–7.25 (m, 2H, Ar-H), 7.46–7.48 (m, 2H, Ar-H), 8.24 (s, 1H, -NH), 8.41–8.44 (m, 1H, -NH), 10.16 (s, 1H, -CHO). ^13^C NMR (100 MHz, CDCl_3_): δ 18.9, 22.6, 30.1, 42.0, 44.7, 45.2, 117.9, 119.4, 127.5, 128.0, 128.6, 129.2, 135.2, 137.3, 139.7, 140.0, 159.9, 162.9, 173.9. MS(ESI): m/z = found 339.5 ([M + H]^+^).

##### 2-(3-benzoylphenyl)-*N*-(4-formamidobenzyl)propenamide (**4b**)

Yield: 75%. White solid. Mp: 95–97 °C. ^1^H NMR (400 MHz, CDCl_3_): δ 1.40 (d, *J* = 8.00 Hz, 3H, -CH_3_), 3.80 (q, *J* = 8.00 Hz, 1H, -CH), 4.19–4.22 (m, 2H, -CH_2_), 7.08–7.10 (m, 2H, Ar-H), 7.48–7.77 (m, 11H, Ar-H), 8.24 (s, 1H, -NH), 8.55–8.58 (m, 1H, -NH), 10.17 (s, 1H, -CHO). ^13^C NMR (100 MHz, CDCl_3_): δ 18.9, 42.1, 45.4, 117.9, 119.5, 128.0, 128.6, 129.0, 130.1, 132.1, 133.2, 135.0, 137.4, 137.5, 143.2, 159.9, 162.9, 173.3, 196.3. MS(ESI): m/z = found 387.5 ([M + H]^+^).

##### *N*-(4-formamidobenzyl)-2-(6-methoxynaphthalen-2-yl)propenamide (**4c**)

Yield: 70%. White solid. Mp: 82–84 °C. ^1^H NMR (400 MHz, DMSO): δ1.44 (d, *J* = 8.00 Hz, 3H, -CH_3_), 3.79 (q, *J* = 8.00 Hz, 1H, -CH), 3.87 (s, 3H, -OCH_3_), 4.19–4.21 (m, 2H, -CH_2_), 7.07–7.16 (m, 4H, Ar-H), 7.29 (s, 1H, Ar-H), 7.46–7.49 (d, 1H, *J* = 8.00 Hz, Ar-H), 7.73–7.79 (m, 3H, Ar-H), 8.24 (s, 1H, -NH), 8.50 (s, 1H, -NH), 10.1 (s, 1H, -CHO). ^13^C NMR (100 MHz, DMSO): δ 19.0, 42.2, 45.5, 55.6, 106.1, 117.9, 119.1, 119.5, 125.8, 126.7, 128.1, 128.7, 129.6, 133.6, 135.2, 137.3, 137.9, 157.5, 159.9, 163.0, 173.8. MS(ESI): m/z = found 363.4 ([M + H]^+^).

##### *N*-(4-formamidobenzyl)-2-((3-(trifluoromethyl)phenyl)amino)benzamide (**4d**)

Yield: 75%. White solid. Mp: 77–79 °C. ^1^H NMR (400 MHz, DMSO): δ 4.34–4.42 (m, 2H, -CH_2_), 6.95–6.99 (m, 1H, Ar-H), 7.13–7.28 (m, 4H, Ar-H), 7.35–7.43 (m, 4H, Ar-H), 7.45–7.47 (m, 1H, Ar-H), 7.52–7.58 (m, 2H, Ar-H), 7.73 (brs, 1H, -NH), 8.25 (brs, 1H, -NH), 9.12 (brs, 1H, -NH), 9.69 (s, 1H, -CHO). ^13^C NMR (100 MHz, DMSO): δ 42.6, 114.6, 117.3, 117.5, 118.0, 119.6, 119.7, 120.4, 121.6, 122.1, 123.3, 126.0, 127.3 (q, *J* = 271 Hz), 128.3, 128.6, 128.9, 129.5, 130.6 (q, *J* = 32 Hz), 130.9, 132.4, 135.0, 137.4, 143.0, 143.6. MS(ESI): m/z = found 414.4 ([M + H]^+^).

##### 2-(2-fluoro-[1,1’-biphenyl]-4-yl)-*N*-(4-formamidobenzyl)propenamide (**4e**)

Yield: 70%. White solid. Mp: 82–84 °C. ^1^H NMR (400 MHz, DMSO): δ1.40 (d, *J* = 8.00 Hz, 3H, -CH_3_), 3.74 (q, *J* = 8.00 Hz, 1H, -CH), 4.23 (s, 2H, -CH_2_), 7.14–7.16 (m, 2H, Ar-H), 7.25–7.27 (m, 2H, Ar-H), 7.39–7.41 (m, 1H, Ar-H), 7.46–7.55 (m, 7H, Ar-H), 8.25 (s, 1H, -NH), 8.54 (s, 1H, -NH), 10.15 (s, 1H, -CHO). ^13^C NMR (100 MHz, DMSO): δ 18.8, 42.3, 45.1, 115.3 (d, *J* = 92 Hz), 118.0, 119.5, 124.3, 126.8 (d, *J* = 28 Hz), 128.2, 128.7, 129.1, 129.2, 131.0, 135.1, 135.5, 137.4, 144.6, 158.1, 160.0, 160.5, 163.0, 173.2. MS(ESI): m/z = found 377.4 ([M + H]^+^).

##### 2-(1,8-diethyl-1,3,4,9-tetrahydropyrano[3,4-b]indol-1-yl)-*N*-(4-formamidobenzyl) acetamide (**4f**)

Yield: 72%. White solid. Mp: 102–104 °C. ^1^H NMR (400 MHz, DMSO): δ 0.66 (t, *J* = 4.00 Hz, 3H, -CH_3_), 1.26 (t, *J* = 4.00 Hz, -CH_3_), 2.02–2.08 (m, 2H, -CH_2_), 2.57–2.70 (m, 2H, CH_2_), 2.72–2.76 (m, 1H, -CH), 2.81.2.87 (m, 2H, -CH_2_), 2.89–2.93 (m, 1H, -CH), 3.96–3.98 (m, 2H, -CH_2_), 4.17–4.32 (m, 2H, -CH_2_), 6.88–6.91 (m, 2H, Ar-H), 7.04–7.16 (m, 2H, Ar-H), 7.23–7.25 (m, 1H, Ar-H), 7.48–7.50 (m, 1H, Ar-H), 8.13–8.15 (m, 1H, -NH), 10.16 (s, 1H, -NH), 10.54 (s, 1H, -CHO). ^13^C NMR (100 MHz, DMSO): δ 8.3, 14.8, 22.4, 24.2, 31.0, 42.1, 44.4, 60.4, 76.0, 107.4, 115.9, 117.9, 119.2, 119.5, 120.1, 126.6, 127.0, 128.0, 128.5, 134.9, 137.2, 159.9, 162.9, 169.8. MS(ESI): m/z = found 420.5 ([M + H]^+^).

##### 2-((2,3-dimethylphenyl)amino)-*N*-(4-formamidobenzyl)benzamide (**4g**)

Yield: 65%. White solid. Mp: 109–110 °C. ^1^H NMR (400 MHz, DMSO): δ 2.10 (s, 3H, -CH_3_), 2.27 (s, 3H, -CH_3_), 4.44–4.45 (m, 2H, -CH_2_), 6.73–6.76 (m, 1H, Ar-H), 6.83–6.86 (m, 1H, Ar-H), 6.92–6.94 (m, 1H, Ar-H), 7.04–7.08 (m, 2H, Ar-H), 7.15–7.27 (m, 2H, Ar-H), 7.29–7.32 (m, 2H, Ar-H), 7.55–7.57 (m, 1H, Ar-H), 7.72–7.74 (m, 1H, Ar-H), 8.26 (s, 1H, -NH), 9.10 (s, 1H, -NH), 9.61(s, 1H, -NH), 10.2 (s, 1H, -CHO). ^13^C NMR (100 MHz, DMSO): δ 14.0, 20.7, 42.5, 114.3, 117.2, 118.1, 119.6, 120.4, 125.6, 126.3, 128.3, 128.8, 129.1, 130.0, 132.5, 135.2, 137.4, 138.2, 139.7, 146.8, 159.9, 163.0, 169.5. MS(ESI): m/z = found 374.5 ([M + H]^+^).

##### *N*-(4-formamidobenzyl)-2-(4-((2-oxocyclopentyl)methyl)phenyl)propenamide (**4h**)

Yield: 68%. White solid. Mp: 107–109 °C. ^1^H NMR (400 MHz, CDCl_3_): δ 1.34 (d, *J* = 8.00 HZ, 3H, -CH_3_), 1.45–1.50(m, 1H, -CH-), 1.65–1.72 (m, 1H, -CH-), 1.83–1.95 (m, 2H, -CH_2_), 1.99–2.11 (m, 1H, -CH-), 2.21–2.28 (m, 1H, -CH-), 2.35–2.46 (m, 2H, -CH_2_), 2.93–2.97 (m, 1H, -CH-), 3.62 (q, *J* = 8.00 Hz, -CH), 4.18–4.20 (m, 2H, -CH_2_), 7.07–7.11 (m, 4H, Ar-H), 7.23–7.25 (m, 2H, Ar-H), 7.47–7,49 (m, 2H, Ar-H), 8.24 (s, 1H, -NH), 8.41–8.44 (m, 1H, -NH), 10.16 (s, 1H, -CHO). ^13^C NMR (100 MHz, CDCl_3_): δ 18.9, 20.5, 29.2, 35.0, 38.1, 42.1, 45.2, 50.5, 117.9, 119.4, 127.7, 128.0, 128.6, 129.0, 135.2, 135.4, 137.4, 138.7, 140.3, 159.9, 162.9, 173.8, 220.3. MS(ESI): m/z = found 379.5 ([M + H]^+^).

#### 4.2.3. General Procedure for the Synthesis of Compounds **5a**–**5h**

A solution of triphosgene (0.5 eq.) in CH_2_Cl_2_ (5 mL) was slowly added to a solution of compound **4** (**a**–**h**) (1.0 eq.), Et_3_N (4.0 equiv) and CH_2_Cl_2_ (5 mL). The resulting mixture was refluxed for 2.0 h in the dark and under a nitrogen atmosphere. Then, selenium (2.0 eq.) was added, and the mixture was refluxed for a further 12 h. TLC showed the reaction was complete. The desired compound was purified by column chromatogragraphy on silica gel.

##### 2-(4-isobutylphenyl)-*N*-(4-isoselenocyanatobenzyl)propenamide (**5a**)

Yield: 80%. White solid. Mp: 82–84 °C. ^1^H NMR (400 MHz, DMSO): δ 0.86 (d, 6H, *J* = 8.00 Hz, 2CH_3_), 1.35 (d, 3H, *J* = 8.00 Hz, -CH_3_), 1.79 (q, 1H, *J* = 8.00 Hz, -CH), 2.41 (d, 2H, *J* = 8.00 Hz, -CH_2_), 3.64 (q, 1H, *J* = 8.00 Hz, -CH), 4.21–4.23 (m, 2H, -CH_2_), 7.08–7.10 (m, 4H, Ar-H), 7.24–7.28 (m, 4H, Ar-H), 10.06 (s, 1H, -NH). ^13^C NMR (100 MHz, DMSO): δ 19.0, 22.7, 30.1, 42.0, 44.7, 45.3, 125.0, 126.6, 127.5, 129.3, 136.9, 138.7, 139.8, 140.0, 173.9. MS(ESI): m/z = found 401.0 ([M + H]^+^).

##### 2-(3-benzoylphenyl)-*N*-(4-isoselenocyanatobenzyl)propenamide (**5b**)

Yield: 78%. White solid. Mp: 88–90 °C. ^1^H NMR (400 MHz, DMSO): δ 1.41 (d, 3H, *J* = 8.00 Hz, -CH_3_), 3.81 (q, 1H, *J* = 8.00 Hz, -CH), 4.21–4.23 (m, 2H, -CH_2_), 7.10–7.12 (m, 1H, Ar-H), 7.26–7.30 (m, 2H, Ar-H), 7.52–7.60 (m, 4H, Ar-H), 7.65–7.68 (m, 2H, Ar-H), 7.73–7.78 (m, 3H, Ar-H), 8.6 (s, 1H, Ar-H), 10.07 (s, 1H, -NH). ^13^C NMR (100 MHz, DMSO): δ 18.9, 42.1, 45.4, 118.5, 125.1, 126.7, 127.6, 128.6, 128.9, 129.0, 130.1, 132.1, 133.2, 136.8, 137.4, 138.8, 140.9, 143.1, 170.8, 173.4, 179.1, 196.3. MS(ESI): m/z = found 448.5 ([M + H]^+^).

##### *N*-(4-isoselenocyanatobenzyl)-2-(6-methoxynaphthalen-2-yl)propenamide (**5c**)

Yield: 82%. White solid. Mp: 76–78 °C. ^1^H NMR (400 MHz, DMSO): δ 1.46 (d, 3H, *J* = 8.00 Hz, -CH_3_), 3.82 (q, 1H, *J* = 8.00 Hz, -CH), 3.87 (s, 3H, -OCH_3_), 4.24–4.26 (m, 2H, -CH_2_), 7.12–7.16 (m, 2H, Ar-H), 7.24–7.29 (m, 2H, Ar-H), 7.48–7.50 (m, 1H, Ar-H), 8.58–8.61 (m, 1H, Ar-H), 10.06 (s, 1H, -NH). ^13^C NMR (100 MHz, DMSO): δ 19.0, 42.2, 45.6, 55.6, 106.2, 119.1, 125.1, 125.8, 126.7, 127.0, 128.1, 128.9, 129.6, 133.6, 137.0, 137.9, 138.7, 141.0, 157.5, 173.9. MS(ESI): m/z = found 424.8 ([M + H]^+^).

##### *N*-(4-isoselenocyanatobenzyl)-2-((3-(trifluoromethyl)phenyl)amino)benzamide (**5d**)

Yield: 80%. White solid. Mp: 100–102 °C. ^1^H NMR (400 MHz, CDCl_3_): δ 4.60 (d, 2H, *J* = 8.00 Hz, -CH_2_), 6.59–6.62 (m, 1H, Ar-H), 6.82–6.86 (m, 1H, Ar-H), 7.22–7.24 (m, 1H, Ar-H), 7.31–7.48 (m, 8H, Ar-H), 9.49 (s, 1H, Ar-H). ^13^C NMR (100 MHz, CDCl_3_): δ 43.2, 116.0, 116.4 (d, *J* = 4 Hz), 118.5, 118.7 (d, *J* = 4 Hz), 119.2, 123.1, 124.0 (q, *J* = 271 Hz), 126.5, 127.6, 128.8, 129.6, 129.9, 131.7 (q, *J* = 32 Hz), 132.8, 138.3, 142.2, 144.6, 169.3. MS (ESI): m/z = found 475.5 ([M + 1]^+^).

##### 2-(2-fluoro-[1,1’-biphenyl]-4-yl)-*N*-(4-isoselenocyanatobenzyl)propenamide (**5e**)

Yield: 82%. White solid. Mp: 88–90 °C. ^1^H NMR (400 MHz, CDCl_3_): δ 1.56 (d, 3H, *J* = 8.00 Hz, -CH_3_), 3.62 (q, 1H, *J* = 8.00 Hz, -CH), 4.38–4.40 (m, 2H, -CH_2_), 5.91–5.94 (m, 1H, Ar-H), 7.10–7.21 (m, 5H, Ar-H), 7.37–7.45 (m, 4H, Ar-H), 7.51–7.54 (m, 2H, Ar-H). ^13^C NMR (100 MHz, CDCl_3_): δ 18.5, 43.0, 46.6, 115.3 (d, *J* = 23 Hz), 123.6 (d, *J* = 4 Hz), 126.4, 127.9, 128.3 (d, *J* = 13Hz), 128.6 (d, *J* = 7 Hz), 128.8, 128.9 (d, *J* = 3 Hz), 129.4 (NCSe), 131.2 (d, *J* = 4 Hz), 135.2, 138.7, 142.4 (d, *J* = 7 Hz), 158.6, 161.1, 173.6. MS (ESI): m/z = found 438.7 ([M + 1]^+^).

##### 2-(1,8-diethyl-1,3,4,9-tetrahydropyrano[3,4-b]indol-1-yl)-*N*-(4-isoselenocyanatobenzyl)acetamide (**5f**)

Yield: 78%. White solid. Mp: 107–109 °C. ^1^H NMR (400 MHz, CDCl_3_): δ 0.87 (t, *J* = 4.00 Hz, 3H, -CH_3_), 1.24 (t, *J* = 4.00 Hz, -CH_3_), 1.85–1.93 (m, 2H, -CH), 2.08–2.14 (m, 1H, -CH), 2.58–2.81 (m, 4H, 2CH_2_), 3.01 (s, 2H, -CH_2_), 4.03–4.09 (m, 2H, -CH_2_), 4.26 (s, 3H, -OCH_3_), 6.80–6.82 (m, 2H, Ar-H), 7.02–7.03 (m, 1H, Ar-H), 7.09–7.13 (m, 1H, Ar-H), 7.25–7.29 (m, 2H, Ar-H), 9.62 (s, 1H, Ar-H). ^13^C NMR (100 MHz, CDCl_3_): δ 7.7, 14.1, 22.2, 24.1, 31.2, 42.5, 43.8, 60.4, 76.0, 107.4, 115.8, 119.8, 120.6, 126.1, 126.3, 127.0, 128.2, 128.3, 129.1, 134.8, 135.5, 138.5, 171.4. MS(ESI): m/z = found 482.1 ([M + 1]^+^).

##### 2-((2,3-dimethylphenyl)amino)-*N*-(4-isoselenocyanatobenzyl)benzamide (**5g**)

Yield: 72%. White solid. Mp: 91–93 °C. ^1^H NMR (400 MHz, DMSO): δ 2.08 (s, 3H, -CH_3_), 2.26 (s, 3H, -CH_3_), 4.51–4.53 (m, 2H, -CH_2_), 6.74–6.75 (m, 1H, Ar-H), 6.83–6.85 (m, 1H, Ar-H), 6.92–6.94 (m, 1H, Ar-H), 7.04–7.09 (m, 2H, Ar-H), 7.45–7.55 (m, 3H, Ar-H), 7.71–7.76 (m, 1H, Ar-H), 9.19 (s, 1H, -NH), 9.58 (s, 1H, -NH). ^13^C NMR (100 MHz, DMSO): δ 14.0, 20.7, 42.4, 114.4, 116.8, 117.2, 119.6, 120.5, 125.7, 126.3, 126.8, 127.0, 128.3, 128.6, 128.9, 129.1, 130.0, 132.7, 138.2, 139.6, 142.2, 147.0, 169.7. MS(ESI): m/z = found 435.1 ([M + Na]^+^).

##### *N*-(4-isoselenocyanatobenzyl)-2-(4-((2-oxocyclopentyl)methyl)phenyl)propenamide (**5h**)

Yield: 75%. White solid. Mp: 86–88 °C. ^1^H NMR (400 MHz, DMSO): δ 1.36 (d, *J* = 8.00 Hz, 3H, -CH_3_), 1.45–1.51 (m, 1H, -CH), 1.66–1.73 (m, 1H, -CH), 1.82–1.93 (m, 2H, -CH_2_), 2.02–2.11 (m, 1H, -CH), 2.21–2.26 (m, 1H, -CH), 2.36–2.46 (m, 2H, -CH), 2.94–2.97 (m, 1H, -CH), 3.62 (q, *J* = 8.00 Hz, 1H, -CH), 4.46 (d, *J* = 8.00 Hz, 2H, -CH_2_), 7.11–7.13(m, 2H, Ar-H), 7.21–7.26 (m, 4H, Ar-H), 7.40–7.42 (m, 2H, Ar-H), 8.52 (s, 1H, -NH). ^13^C NMR (100 MHz, DMSO): δ 18.9, 20.5, 29.2, 35.0, 38.1, 42.0, 45.3, 50.5, 125.1, 126.6, 127.7, 128.1, 128.8, 129.1, 136.9, 138.8, 140.2, 140.3, 141.1, 173.9, 174.0, 179.0. MS(ESI): m/z = found 440.5 ([M + Na]^+^).

#### 4.2.4. General Procedure for the Synthesis of Compounds **6a**–**6h**

Compound **5** (**a**–**h**) (1.0 eq.) was dissolved in CH_3_OH (20 mL) at room temperature, then the solvent was refluxed for 4.0 h. TLC showed the reaction was complete. The desire compound was purified by column chromatogragraphy on silica gel.

##### *O*-methyl (4-((2-(4-isobutylphenyl)propanamido)methyl)phenyl) carbamoselenoate (**6a**)

Yield: 60%. White solid. Mp: 117–118 °C. ^1^H NMR (400 MHz, CDCl_3_): δ 0.88 (d, *J* = 8.00 Hz, 6H, 2CH_3_), 1.44 (d, 3H, *J* = 8.00 Hz, -CH_3_), 1.83 (q, 1H, *J* = 8.00 Hz, -CH), 2.44 (d, 2H, *J* = 8.00 Hz, -CH_2_), 3.64 (q, 1H, *J* = 8.00 Hz, -CH), 4.12 (s, 3H, -OCH_3_), 4.23–4.34 (m, 2H, -CH_2_), 7.07–7.12 (m, 4H, Ar-H), 7.18–7.24 (m, 4H, Ar-H), 7.51 (brs, 1H, -NH). ^13^C NMR (100 MHz, CDCl_3_): δ 17.3, 21.3, 30.1, 42.1, 44.6, 45.8, 60.7, 121.8, 126.8, 127.5, 128.9, 136.1, 136.6, 139.0, 140.2, 175.8, 191.8. MS(ESI): m/z = found 455.1 ([M + Na]^+^).

##### *O*-methyl (4-((2-(3-benzoylphenyl)propanamido)methyl)phenyl)carbamoselenoate (**6b**)

Yield: 62%. White solid. Mp: 88–90 °C. ^1^H NMR (400 MHz, CDCl_3_): δ 1.48 (d, *J* = 8.00 Hz, 3H, -CH_3_), 3.77 (q, *J* = 8.00 Hz, 1H, -CH), 4.12 (s, 3H, -OCH_3_), 4.25–4.35 (m, 2H, -CH_2_), 7.11–7.18 (m, 3H, Ar-H), 7.45–7.51 (m, 4H, Ar-H), 7.60–7.64 (m, 3H, Ar-H), 7.72–7.76 (m, 3H, Ar-H). ^13^C NMR (100 MHz, CDCl_3_): δ 17.4, 42.2, 45.9, 60.7, 121.8, 124.1, 127.7, 128.2, 128.3, 128.7, 129.7, 131.4, 132.5, 135.9, 136.6, 137.3, 137.7, 142.2, 174.9, 191.8, 197.1. MS (ESI): m/z = found 503.0 ([M + Na]^+^).

##### *O*-methyl (4-((2-(6-methoxynaphthalen-2-yl)propanamido)methyl)phenyl)carbamoselenoate (**6c**)

Yield: 55%. White solid. Mp: 93–95 °C. ^1^H NMR (400 MHz, CDCl_3_): δ 1.52 (d, *J* = 8.00 Hz, 3H, -CH_3_), 3.79 (q, *J* = 8.00 Hz, 1H, -CH), 3.87 (s, 3H, -OCH_3_), 4.12 (s, 3H, -OCH_3_), 4.26–4.30 (m, 2H, -CH_2_), 7.08–7.17 (m, 5H, Ar-H), 7.40–7.43 (m, 1H, Ar-H), 7.65–7.71 (m, 3H, Ar-H). ^13^C NMR (100 MHz, CDCl_3_): δ 17.4, 42.2, 46.1, 54.4, 60.7, 105.2, 118.5, 121.7, 124.1, 125.4, 125.8, 126.8, 127.6, 128.8, 129.0, 133.8, 136.1, 136.7, 157.7, 175.7, 191.7. MS(ESI): m/z = found 479.1 ([M + Na]^+^).

##### *O*-methyl (4-((2-((3-(trifluoromethyl)phenyl)amino)benzamido)methyl)phenyl)carbamoselenoate (**6d**)

Yield: 65%. White solid. Mp: 101–102 °C. ^1^H NMR (400 MHz, CDCl_3_): δ 4.12 (s, 3H, -OCH_3_), 4.46–4.48 (m, 2H, -CH_2_), 6.89–6.93 (m, 1H, Ar-H), 7.14–7.16 (m, 1H, Ar-H), 7.21–7.38 (m, 9H, Ar-H), 7.62–7.65 (m, 1H, Ar-H). ^13^C NMR (100 MHz, CDCl_3_): δ 42.5, 64.8, 114.4 (*J* = 4.0 Hz), 116.9, 117.2 (*J* = 4.0 Hz), 120.0, 121.3, 121.5, 121.9, 122.9, 125.6, 127.8, 128.6, 129.8, 131.2 (q, *J* = 32 Hz, -CF_3_), 131.9, 136.29 (d, *J* = 58 Hz), 143.1 (d, *J* = 32 Hz). MS (ESI): m/z = found 530.1 ([M + Na]^+^).

##### *O*-methyl (4-((2-(2-fluoro-[1,1’-biphenyl]-4-yl)propanamido)methyl)phenyl)carbamoselenoate (**6e**)

Yield: 60%. White solid. Mp: 72–74 °C. ^1^H NMR (400 MHz, CDCl_3_): δ 1.56 (d, 3H, *J* = 8.00 Hz, -CH_3_), 3.62 (q, *J* = 8.00 Hz, 1H, -CH), 4.21 (s, 3H, -OCH_3_), 4.35–4.40 (m, 2H, -CH_2_), 5.91–5.97 (m, 1H, -NH), 7.10–7.18 (m, 6H, Ar-H), 7.37–7.53 (m, 6H, Ar-H), 9.22 (brs, 1H, -NH). ^13^C NMR (100 MHz, CDCl_3_): δ 18.8, 42.2, 45.1, 61.6, 115.3 (d, *J* = 23 Hz), 122.6, 124.3, 125.2, 126.7, 127.0, 127.7, 128.1 (d, *J* = 7 Hz), 128.9, 129.1, 129.2, 130.9 (d, *J* = 3 Hz), 135.5, 136.8 (d, *J* = 19 Hz), 140.9, 144.6, 158.1, 160.5, 173.2 (d, *J* = 16 Hz), 190.7. MS (ESI): m/z = found 493.0 ([M + Na]^+^).

##### *O*-methyl (4-((2-(1,8-diethyl-1,3,4,9-tetrahydropyrano[3,4-b]indol-1-yl)acetamido)methyl)phenyl)carbamoselenoate (**6f**)

Yield: 62%. White solid. Mp: 83–85 °C. ^1^H NMR (400 MHz, CDCl_3_): δ 0.87 (t, *J* = 4.00 Hz, 3H, -CH_3_), 1.24 (t, *J* = 4.00 Hz, -CH_3_), 1.87–2.14 (m, 2H, -CH_2_), 2.63–2.80 (m, 4H, 2CH_2_), 3.03 (s, 2H, -CH_2_), 4.03–4.07 (m, 2H, -CH_2_), 4.26 (s, 3H, -OCH_3_), 6.78–6.85 (m, 3H, Ar-H), 7.00–7.13 (m, 2H, Ar-H), 7.24–7.29 (m, 2H, Ar-H). ^13^C NMR (100 MHz, CDCl_3_): δ 8.3, 14.9, 22.4, 24.2, 31.1, 42.0, 44.3, 60.4, 61.7, 76.0, 107.4, 115.9, 119.2, 120.1, 122.5, 126.5, 127.0, 127.9, 128.7, 134.9, 137.0, 141.0, 169.9, 190.6. MS(ESI): m/z = found 536.1 ([M + Na]^+^).

##### *O*-methyl (4-((2-((2,3-dimethylphenyl)amino)benzamido)methyl)phenyl)carbamoselenoate (**6g**)

Yield: 62%. White solid. Mp: 83–85 °C. ^1^H NMR (400 MHz, CDCl_3_): δ 2.13 (s, 3H, -CH_3_), 2.28 (s, 3H, -CH_3_), 4.15 (s, 3H, -OCH_3_), 4.50 (s, 2H, -CH_2_), 6.69–6.73 (m, 1H, Ar-H), 6.79–6.81 (m, 1H, Ar-H), 6.91–6.93 (m, 1H, Ar-H), 7.02–7.05 (m, 2H, Ar-H). 7.17–7.21 (m, 1H, Ar-H), 7.25–7.32 (m, 4H, Ar-H), 7.57–7.61 (m, 1H, Ar-H). ^13^C NMR (100 MHz, CDCl_3_): δ 12.6, 19.3, 42.3, 60.7, 114.3, 116.8, 117.3, 120.6, 121.9, 125.4, 125.6, 127.7, 128.1, 130.3, 131.9, 136.3, 136.6, 137.7, 139.5, 146.8, 170.5, 191.8. MS (ESI): m/z = found 490.1 ([M + Na]^+^).

##### *O*-methyl (4-((2-(4-((2-oxocyclopentyl)methyl)phenyl)propanamido)methyl)phenyl)carbamoselenoate (**6h**)

Yield: 68%. White solid. Mp: 109–111 °C. ^1^H NMR (400 MHz, CD_3_OD): 1.34 (d, 3H, *J* = 8.00 Hz, -CH_3_), 1.45–1.50 (m, 1H, -CH-), 1.67–1.72 (m, 1H, -CH-), 1.83–1.92 (m, 2H, -CH_2_), 2.02–2.11(m, 1H, -CH-), 2.21–2.26 (m, 1H, -CH-), 2.36–2.44 (m, 2H, -CH_2_), 2.94–2.97 (m, 1H, -CH-), 3.62–3.63 (m, 1H, -CH-), 4.09 (m, 3H, -OCH_3_), 4.18 (s, 2H, -CH_2_), 7.11–7.13 (m, 4H, Ar-H), 7.23–7.27 (m, 4H, Ar-H), 8.44 (s, 1H, -NH), 11.7 (s. 1H, -NH). ^13^C NMR (100 MHz, CD_3_OD): 18.9, 20.5, 29.2, 35.0, 38.1, 42.1, 45.2, 50.5, 61.6. 122.6, 124.7, 127.7, 128.0, 129.0, 136.7, 137.1, 138.7, 140.3, 173.9, 190.7. MS (ESI): m/z = found 495.0 ([M + H]^+^).

### 4.3. Cell Lines and Growth Conditions

Exponentially growing cells were harvested and plated on 96-well plates at a concentration of 1 × 104 cells/well. After 24 h incubation at 37 °C under a humidified 5% CO_2_ to allow cell attachment, the cells in the wells were treated with target compounds at various concentrations for 48 h. The concentration of DMSO was always kept below 1.25%, which was found to be non-toxic to the cells. Three hours prior to experiment termination, MTT solution (20 μL of 5.0 mg/mL solution) was added to each well and incubated at 37 °C. At the termination timepoint, the medium/MTT mixtures were removed, and the formazan crystals formed by the mitochondrial dehydrogenase activity of vital cells were dissolved in 100 μL of DMSO per well. The optical densities were measured at 570 nm using a 96-well multiscanner (Dynex Technologies, MRX Revelation; Chantilly, VA, USA).

### 4.4. Detection of Bcl-2, IL-2 and Caspase-3 Molecular Biomarkers in MCF-7 Cells

Bcl-2, IL-2 and caspase-3 cells were evaluated in MCF-7 cells treated with the corresponding target compounds and incubated for 48 h and compared with their levels in control, untreated MCF-7 cell line. The cells were harvested by trypsinization and lysed by lysate buffer (Beyotime Biotech, Nanjing, China). Protein levels of Bcl-2, IL-2 and caspase-3 were measured using enzyme-linked immunosorbent assay (ELISA) by multifunctional enzyme marker (Molecular Devices i3, San Jose, CA, USA) at a wavelength of 570 nm.

### 4.5. DPPH Free Radical Scavenging Activity

DPPH free radical scavenging activity of corresponding compounds was measured according to the previously reported method with little optimization. In brief, 20 mL of test samples at different concentrations were mixed with 180 mL of or DPPH solution for 30 min in the dark. Then, the change in absorbance at 517 nm for DPPH was measured on a microplate reader. Ascorbic acid (vitamin C) and ebselen were used as a positive control; DMSO was used as a negative control.

### 4.6. Bleomycin-Dependent DNA Damage

The reaction mixture contained DNA (0.5 mg/mL), bleomycin sulfate (0.05 mg/mL), MgCl_2_ (5 mM), FeCl_3_ (50 mM), and tested compound in a conc. of 0.1 mg/mL. L-ascorbic acid was used as positive control. The mixture was incubated at 37 °C for 1 h. The reaction was terminated by addition of 0.05 mL EDTA (0.1 M). The color was developed by adding 0.5 mL TBA (1% *w*/*v*) and 0.5 mL HCl (25% *v*/*v*), followed by heating at 80 °C for 30 min. After cooling in ice water, the extent of DNA damage was measured by the increase in absorbance at 532 nm.

### 4.7. Molecular Modeling

#### 4.7.1. Protein and Ligand Preparation

The mammalian TrxR1 protein (PDB ID: 1H6V) used for docking was obtained from Protein Data Bank. The original structure was prepared using Protein Preparation Wizard in Maestro 11.5 (Schrödinger Release 2018-1: Maestro, Schrödinger, LLC, New York, NY, USA, 2018), with all but one subunit (E) discarded, bond orders assigned, hydrogens added, ionization and tautomerization state adjusted, hydrogen bond assignment optimized, waters removed, and structure minimized.

The LigPrep utility in Maestro 11.5 was used to perform ligand preparation, applying an OPLS3 force field. The generation of tautomers and possible ionization states was mediated by Epik utility. All stereoisomers were considered to be generated, followed by minimization of the resulting 3D conformations. There was no filtration process during preparation.

#### 4.7.2. Ligand Docking

The docking task was carried out in Discovery Studio 2018 (Dassault Systèmes BIOVIA, Discovery Studio 2018, San Diego: Dassault Systèmes, 2018). The prepared TrxR1 protein was typed in CHARMm forcefield and the docking site was defined as a sphere with center coordinates X: 27.757, Y: 6.510, Z: 33.698 and a radius of 15 Å. Using Flexible Docking protocol, the residue sidechains within the site sphere were allowed to move. Ten protein conformations were created, with a maximum alteration of 8 residues. The FAST method was adopted; up to 25 conformations per ligand were generated with an energy threshold of 20 kcal. With all other parameters as default, ligands were preliminarily docked into each protein structure. After the removal of similar poses by clustering, the remaining complexes were refined and minimized, leading to a total of 133 final poses.

#### 4.7.3. Result Analysis

The resulting 133 poses were clustered by ligand (53 for 3 h, 40 each for 2 h and 3 h) and visualized in Maestro 11.5. For each of the poses, the distance between the compound’s selenium atom and the sulfur atom of either Cys497 or Cys498 was calculated as a measurement of covalent bonding probability. Any complex with less than 5 Å of the distance above was counted potentially reactive. For each ligand, average –CDocker energy and average selenium–sulfur distance were calculated; the latter was –CDocker energy weighted.

## Data Availability

Data are available from the corresponding authors upon reasonable request. ^1^H, ^13^C NMR and MS spectra of compounds are available from the Appendix A.

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
