# Peer review of "New Organoselenium (NSAIDs-Selenourea and Isoselenocyanate) Derivatives as Potential Antiproliferative Agents: Synthesis, Biological Evaluation and in Silico Calculations"

_molecules, 2022, doi:10.3390/molecules27144328_

Round 1
Reviewer 1 Report
The authors synthesized several organoselenium derivatives and set out to investigate their DPPH-scavenging, DNA-protecting (against bleomycin), GPX-like, and antiproliferative activities. Antiproliferative assays were conducted using exclusively cancerous cell lines (HeLa, MCF-7, SW480, and A549). Moreover, the authors also investigated the effect of a couple of these compounds on apoptosis markers (Bcl-2, IL-2, and Caspase-3) in MCF-7 cells. Lastly, in silico simulations were conducted to better understand possible mechanisms underlying the effects of these two compounds (6b and 6f). The introduction provide enough background information and states the purpose of the study. The methodology is scientifically sound (although I cannot comment on the synthetic routes used). The results are logically presented, but some of them lack appropriate statistical analysis. Conclusions are supported by data. However, the significance of the results is highly limited because of the lack of results using non-cancerous cell lines (i.e., “normal” cells). In other words, we do not know the effect of these compounds on “healthy” normal cells. Thus, it might not be a great result if the substances impair the homeostasis of both cancerous and non-cancerous cells. This is absolutely necessary to call something “anti-cancer”, and not simply cytotoxic. To sum up, the authors should present data for non-cancerous cells or acknowledge the lack of these data as a major limitation of the study (in a limitations section of the discussion). Here are some specific concerns:
1. Title and elsewhere: There is no data on the selectivity of these compounds towards cancerous cells, thus the term “anticancer” should be replaced by a more suitable term, such as “antiproliferative”. This should be the case unless the authors provide evidence that these compounds do not affect “normal” cells.
2. Line 116: Please include a brief sentence explaining which method was used to measure protein expression. Western blot? ELISA? mRNA levels?
3. Figures 3 and 4: These data must be analyzed statistically to provide evidence to support any statement describing downregulation or upregulation, higher or lower, of these markers.
4. Line 195: It would be useful to provide a brief explanation of why TrxR1 was chosen for the docking simulations.
5. Lines 196-213: These two paragraphs are just methodology and should not be in the results section.
Author Response
Q1. Title and elsewhere: There is no data on the selectivity of these compounds towards cancerous cells, thus the term “anticancer” should be replaced by a more suitable term, such as “antiproliferative”. This should be the case unless the authors provide evidence that these compounds do not affect “normal” cells.
A1,We have replace “anticancer” to “antiproliferative” in the manuscript.
Q2. Line 116: Please include a brief sentence explaining which method was used to measure protein expression. Western blot? ELISA? mRNA levels?
A2, We use ELISA to measure protein expression and explain in the manuscript.
Q3. Figures 3 and 4: These data must be analyzed statistically to provide evidence to support any statement describing downregulation or upregulation, higher or lower, of these markers.
A3, We provide tables to support our describe.
Q4. Line 195: It would be useful to provide a brief explanation of why TrxR1 was chosen for the docking simulations.
A4, We give our explanation of why we choose TrxR1 as docking protein in 2.5 section
Reviewer 2 Report
My only concern, however, is that many of the reported compounds are not chemically pure. And this point should be clarified before publication. My guess is based on the analysis of the 13C-NMR spectra from the SI section. For the aromatic region, much more signals are seen than expected for a given structure. See the spectra recorded for 4e, 4g and 5a as examples
Author Response
Q1, My only concern, however, is that many of the reported compounds are not chemically pure. And this point should be clarified before publication. My guess is based on the analysis of the 13C-NMR spectra from the SI section. For the aromatic region, much more signals are seen than expected for a given structure. See the spectra recorded for 4e, 4g and 5a as examples
A1, The compound obtained by silico chromatography, the purity of compounds in this manuscript are >95%.
Round 2
Reviewer 1 Report
I have no further comments.
Author Response
I have no further reply
Reviewer 2 Report
Please explain why::
for 4e you observe 31 signals instead 25
for 4g you observr 31 signals instead 23
for 5a you observr 13 signals instead 8 ( aromatic region)
Q1, My only concern, however, is that many of the reported compounds are not chemically pure. And this point should be clarified before publication. My guess is based on the analysis of the 13C-NMR spectra from the SI section. For the aromatic region, much more signals are seen than expected for a given structure. See the spectra recorded for 4e, 4g and 5a as examples
A1, The compound obtained by silico chromatography, the purity of compounds in this manuscript are >95%.
Author Response
For 4e, the compound include F atom, which make carbon atom split in 13C NMR.
For 4g, the compound is not pure enough to make the 13C NMR clearly, if possible, we will repurify this compound.
For 5a, the aromatic region of carbon signal is 8, we delete some impurity peaks.